# Inhibitory Mechanism of Quercimeritrin as a Novel α-Glucosidase Selective Inhibitor

**DOI:** 10.3390/foods12183415

**Published:** 2023-09-13

**Authors:** Fengyu Guo, Jie An, Minlong Wang, Weibo Zhang, Chong Chen, Xueying Mao, Siyuan Liu, Pengjie Wang, Fazheng Ren

**Affiliations:** 1College of Food Science and Nutritional Engineering, China Agricultural University, Beijing 100083, China; guofengyu1116@163.com (F.G.); maoxueying@cau.edu.cn (X.M.); 2Department of Nutrition and Health, China Agricultural University, Beijing 100083, China; jie_an@cau.edu.cn (J.A.); mlwang@cau.edu.cn (M.W.); zhangweibo@cau.edu.cn (W.Z.); chenchong409@cau.edu.cn (C.C.); siyuan.liu@cau.edu.cn (S.L.); 3Food Laboratory of Zhongyuan, China Agricultural University, Beijing 100083, China

**Keywords:** flavonoid glycosides, α-glucosidase, α-amylase, selective inhibitor

## Abstract

In this study, 12 flavonoid glycosides were selected based on virtual screening and the literature, and Quercimeritrin was selected as the best selective inhibitor of α-glucosidase through in vitro enzyme activity inhibition experiments. Its IC_50_ value for α-glucosidase was 79.88 µM, and its IC_50_ value for α-amylase >250 µM. As such, it could be used as a new selective inhibitor of α-glucosidase. The selective inhibition mechanism of Quercimeritrin on the two starch-digesting enzymes was further explored, and it was confirmed that Quercimeritrin had a strong binding affinity for α-glucosidase and occupied the binding pocket of α-glucosidase through non-covalent binding. Subsequently, animal experiments demonstrated that Quercimeritrin can effectively control postprandial blood glucose in vivo, with the same inhibitory effect as acarbose but without side effects. Our results, therefore, provide insights into how flavone aglycones can be used to effectively control the rate of digestion to improve postprandial blood glucose levels.

## 1. Introduction

In the treatment of obesity and diabetes, the control of postprandial blood glucose has attracted widespread attention, given the close link between postprandial blood glucose and diabetic complications [1]. The control of postprandial blood glucose mainly involves controlling starch-digesting enzymes, which mainly include salivary amylase, an amylase secreted by the pancreas, and α-glucosidase [2,3]. Currently, clinically used drugs (including acarbose, voglibose, and miglitol) mainly inhibit α-amylase and α-glucosidase [4,5]. However, it is precisely because of the severe inhibition of these two enzymes that large amounts of starch are not digested. Most undigested starch can lead to a wide range of gastrointestinal side effects, such as intestinal bloating caused by microbial fermentation of starch in the colon [6]. Therefore, to enhance the application of starch-digesting-enzyme inhibitors and improve the medication experience of patients, the development of new inhibitors must address the serious drawbacks of current clinical drugs [7].

During starch digestion, α-amylase cleaves starch molecules into small maltose oligosaccharides, whereas α-glucosidase digests α-amylase products into glucose as the final digestive product [8]. To solve the side effects of current starch-digesting-enzyme inhibitors and ensure the effective control of postprandial blood glucose, we adopted the method of regulating the inhibitory effect of inhibitors on two digestive enzymes so that the inhibitors selectively inhibit α-glucosidase [9]. This allows the starch to be digested slowly but completely before it enters the colon, thus avoiding the side effects of large amounts of undigested starch. At the same time, it also helps control the sharp rise in blood sugar after meals.

At present, some compounds with selective inhibitory activity toward α-glucosidase have been reported (such as apigenin and quercetin), but the obtained compounds generally have a weak inhibitory effect on α-glucosidase selection [10,11]. Flavonoid glycosides are a class of natural organic compounds widely found in nature, belonging to flavonoids [12]. Flavonoids mostly exist in plants in the form of glycosides. Flavonoid glycosides are widely distributed in nature and have a series of important physiological activities, including inhibition of starch digestion enzymes [13]. Hua et al. found by studying the inhibitory effect of flavonoids on starch-digesting enzymes in Lu’an GuaPian tea that kaempferol monoglycoside can selectively inhibit α-glucosidase [14]. Flavonoid glycosides can bind noncovalently to active site residues of enzymes and have variable structural properties, so they have the potential for the selective inhibition of α-glucosidase [14]. In addition, the literature has shown that extracts of Astragalus creticus, containing quercimeritrin among other secondary metabolites, demonstrated antioxidant potential and good inhibition of α-glucosidase enzyme [15]. Recent studies showed that other flavones, tricetin and genkwanin, exhibited α-glucosidase and α-amylase inhibitory activities, respectively [16,17], while linarin could prevent the action of α-glucosidase and α-amylase enzymes [18]. However, the existence of more potent selective inhibitors of flavone glycosides and the structural requirements for selective inhibition of α-amylase and α-glucosidase by flavonoid glycosides are unclear. We are currently only able to summarize some basic structural requirements from studies investigating the inhibition of α-amylase or α-glucosidase by flavone glycosides alone. It is known that the presence of theOH group at the A5 position and the double bond between C2 and C3 of the C ring are the structural basis for inhibiting α-glucosidase, and that the substitution of different glycosidic groups on the AB ring has different effects on the inhibition of α-amylase and α-glucosidase [10,19].

Therefore, we adopted virtual docking technology and combined it with previous studies to screen out flavonoid glycosides with inhibitory potential. Through experimental screening, we expected to find the best selective inhibitor of α-glucosidase. Exploring the structural characteristics of α-glucosidase selective inhibitors and the inhibitory mechanism of selective inhibition can guide the search for more extensive α-glucosidase selective inhibitors. Our findings provide insights into how flavonoid glycosides can be applied to eliminate existing inhibitor side effects and effectively control postprandial blood sugar.

## 2. Materials and Methods

### 2.1. Materials

Isovitexin, Trifolin, Avicularin, Apigetrin, Quercetagitrin, Epicatechin gallate, Catechin gallate, Quercimeritrin, Quercituron, Quercitrin, Hyperoside, and Oroxin A were obtained from Purify Inc. (Chengdu, China), and the α-glucosidase (EC 232-604-7, lyophilized powder, ≥100 units/mg protein) isolated from *Saccharomyces cerevisiae* and α-amylase (EC 232-565-6, powder, ≥5 units/mg solid) isolated from porcine pancreas were purchased from Sigma-Aldrich Co. (St. Louis, MI, USA). The other reagents used in this experiment are analytically pure reagents purchased from Sigma-Aldrich Co. (St. Louis, MI, USA) unless specified otherwise.

### 2.2. Virtual Screening

Autodock Vina was used to conduct virtual screening of the self-built flavonoid glycosides’ library.

Additionally, 2QMJ and 1CXW were selected from the protein database as the crystal structures for α-glucosidase and α-amylase docking. The scoring results were obtained by docking the compounds with α-glucosidase and α-amylase, respectively, and the compounds were sorted and selected.

### 2.3. In Vitro α-Glucosidase Activity Assay

The in vitro α-glucosidase activity assay was determined using spectrophotometry, as previously reported, with only minor modifications [20]. In phosphate buffer with a pH of 6.8, 0.5 U/mL of α-glucosidase and the appropriate quantity of *p*-nitrophenyl-d-glucopyranoside (pNGP) were produced. All test chemicals, including the standard drug acarbose, were dissolved in DMSO to form a 10 mM mother liquor. Gradient dilution with a phosphate-buffered solution was used to create sample solutions of various concentrations. First, different concentrations of compound (10 μL), an enzyme solution (40 μL), and potassium phosphate buffer (100 μL) were pre-incubated in 96-well plates at 37 °C for 10 min. Then, 50 µL of the substrate (pNGP, 0.6 mM) was added to each microwell, and incubated at 37 °C for 20 min, and the change in enzyme activity was detected by measuring the absorbance at 405 nm. Acarbose and DMSO were utilized as standard and control inhibitors, respectively.

The enzymatic inhibitory activity of the tested compounds was calculated using the following formula:Inhibition %=[(Abs control−Abs sample)/Abs control]×100

The IC_50_ values of the tested compounds were calculated using nonlinear fitting (logit method).

### 2.4. In Vitro α-Amylase Activity Assay

The in vitro α-glucosidase activity assay was determined using spectrophotometry, as previously reported, with only minor modifications [21]. A total of 5 U/mL of α-amylase and the appropriate quantity of 2-chloro-4-nitrophenyl α-d-maltotrioside (G3-CNP) were produced. All test chemicals, including the standard drug acarbose, were dissolved in DMSO to form 10 mM mother liquor. Gradient dilution with a phosphate-buffered solution was used to create sample solutions of various concentrations. First, different concentrations of compound (10 μL), an enzyme solution (40 μL), and potassium phosphate buffer (100 μL) were pre-incubated in 96-well plates at 37 °C for 10 min. Then, 50 µL of the substrate (pNαGP, 0.6 mM) was added to each microwell and incubated at 37 °C for 20 min, and the change in enzyme activity was detected by measuring the absorbance at 405 nm. Acarbose and DMSO were utilized as standard and control inhibitors, respectively.

The enzymatic inhibitory activity of the tested compounds was calculated using the following formula:Inhibition %=[(Abs control−Abs sample)/Abs control]×100

The IC_50_ values of the tested compounds were calculated using nonlinear fitting (logit method).

### 2.5. Inhibition Kinetic Analysis

To further investigate the inhibitory kinetics of Quercimeritrin, α-glucosidase and α-amylase activities were determined at different substrate concentrations [20]. The reactions of a series of compounds with substrates pNαPG and G3-CNP were tested using the Lineweaver–Burk expression in Equation (1) to calculate the maximum velocity (*V*max) and Michaelis–Menten constant (*K*m) values [22].
(1)$$\frac{1}{v}=\frac{Km}{Vmax}\frac{1}{\left[S\right]}+\frac{1}{Vmax},$$
where *V*max and *K*m are the maximum velocity of the enzyme and the Michaelis–Menten constant without inhibitor, respectively. [S] is the concentrations of the substrate and inhibitor.

### 2.6. Fluorescence Quenching Experiment

Based on the previously described techniques, the fluorescence quenching experiment was somewhat modified [23]. Quercimeritrin solutions of various concentrations (1.0 mL, 0–200 mM) were used to titrate the α-glucosidase (1.0 mL, 2 U/mL) and α-amylase (1.0 mL, 2 U/mL), which were then left to equilibrate for 5 min before fluorescence measurement. The reaction solution’s fluorescence intensity was measured in a quartz colorimetric dish using a fluorescence spectrometer (F-7100, Tokyo, Japan) at wavelengths ranging from 295 to 500 nm (emission wavelength) and 280 nm (excitation wavelength) at various temperatures (305.15, 310.15, and 315.15 K) (1.0 cm path length). The 5.0 nm excitation and emission slit widths were predetermined.

At the same time, the changes of microenvironment of fluorophore of tyrosine (Tyr) and tryptophan (Trp) residues were detected. The synchronous fluorescence spectra of 260–320 nm were determined by setting the excitation and emission wavelength intervals to 15 nm and 60 nm, respectively.

### 2.7. Molecular Docking Simulation

In this experiment, AutoDock Vina was used for molecular docking. The first step was to prepare the protein crystal structure. Additionally, 2QMJ and 1CXW were selected from the protein database as the crystal structures for α-glucosidase and α-amylase docking, and the protein crystal structure was treated by removing water molecules and adding hydrogen atoms. The second step is small molecule preparation. ChemBiodraw Ultra 14.0 (Waltham, MA, USA) was used to map the three-dimensional structure of quercetin and minimize the energy of the small molecule. In the third step, a docking box is generated for the center, and AutoDock Vina is used for docking. Ten posturing positions were generated during the docking process, and the posturing with the highest Glide score was selected to study the interaction between quercetin, α-glucosidase, and α-amylase.

### 2.8. Molecular Dynamics Study

Molecular dynamics (MD) simulation was performed in the Yinfo cloud computing platform utilizing AmberTools 20 packets (https://cloud.yinfotek.com/ (accessed on 1 September 2023). The force fields of AMBER ff19SB and GAFF [24] were applied to α-glucosidase proteins and compounds, respectively. The OPC water model employs a truncated octahedral water box with a border of 10 Å. A periodic boundary was used to simulate a neutralized net charge of 0.15 M NaCl in a solvent environment. Then, to eliminate false atomic connections, two steepest dips of class 10,000 and 10,000 and conjugate gradient dips were produced. Following initial optimization, the system was able to integrate 200 ps NPT and 1 ns NPT with balance. Utilizing Langevin dynamics next, the temperature was kept at 300 K, the collision frequency at 1 ps^–1^, the Monte Carlo barometric regulator at 1 ps, and the pressure at 1 atm. In the end, 40 ns MD was created in NVT integration without limitations. The trajectory was examined using the CPPTRAJ module [25].

### 2.9. Postprandial Blood Glucose Level Measurement

db/db mice and C57BL/6J mice were purchased f0rom Beijing Zhishan Health Medical Research Institute. The animal testing procedure was approved by Beijing Huayuan Times Technology Co., Ltd Ethics Committee on the Management and Welfare of Laboratory Animals. (Ethics review batch number: HYSD2023-04) and strictly follows the local and national codes of ethics. All mice were placed in a control chamber at 25 °C and 60% relative humidity with a light/dark period of 12/12 h. During this period, all mice had free access to normal feed and water. After 1 week of feeding, db/db mice were randomly divided into four groups (10 mice per group): normal saline (model group, model), 100 mg/kg acarbose (positive control group, PC), 100 mg/kg Quercimeritrin (low-dose group, LD), 200 mg/kg Quercimeritrin (high-dose group, HD), and C57BL/6J was used as blank controls (normal). After fasting for one night, the BC group, PC group, HD group, LD group, and blank control group were given 10 mL normal saline/kg, 100 mg acarbose normal saline/kg, 100 mg Quercimeritrin normal saline/kg, and 200 mg Quercimeritrin normal saline/kg, respectively. All mice were given starch (2 g/kg) orally 60 min after treatment [21]. Blood samples were taken from the caudal vein and blood glucose was measured after 0, 30, 60, and 120 min. The area under the curve (AUC) was calculated using the trapezoidal rule.

### 2.10. Statistical Analysis

All the results of this experiment were performed three times, and the results were expressed as the mean ± SD. The data were analyzed using one-way analysis of variance using GraphPad Prism 9.0 software. *p* < 0.05 was considered statistically significant.

## 3. Results and Discussion 

### 3.1. Structural Properties of Compounds That Selectively Inhibit Starch-Digesting Enzymes

In this paper, the selective inhibition structural characteristics of 12 flavone aglycones on two starch-digesting enzymes were studied. The basic structural characteristics of flavone aglycones are shown in the upper left corner of Figure 1, mainly the glycosylated polyhydroxy or polymethoxy derivatives of 2-phenyl benzopyrane. The differences in geometry and chemical properties of flavonoid glycosides lead to different inhibitory effects and selectivity on α-glucosidase and α-amylase [26]. The active pocket of α-glucosidase presents a “narrow and deep” shape, mainly acting on the −1, +1 glycan site, whereas the active pocket of α-amylase presents a “wide and shallow” shape, and the active pocket of α-amylase is larger, mainly acting on a total of 5 glycan sites from −3 to 2. Thus, there are differences between the two enzymes in terms of the geometry of the active pocket [27]. In addition, there are differences in the chemical properties of α-glucosidase and α-amylase active pockets, which are the basis for our selective inhibition of enzymes through the adjustment of the structural properties of compounds [28].

Figure 1 shows the maximum inhibitory properties (%) of the 12 different flavonoid glycoside structures on the two starch-digesting enzymes at a concentration of 500 µM. Quercimeritrin and Hyperoside have the same molecular weight; the difference is only that the glycoside group is transferred from the A7 position to the C3 position, resulting in a decrease in the inhibitory activity of Hyperoside in both enzymes, indicating that the glycosylation at the C3 position is not conducive to the inhibition of the two enzymes, whereas glycosylation at the A7 position is more conducive to the improvement of enzyme inhibitory activity than OH. By measuring kaempferol 3-O-β-d-glucosyl-(1-2)-b-d-galactoside and rhamnetin, Milella et al. found that the glycan group at the 3-position of the C ring is not conducive to the inhibition of α-amylase compared with theOH group [29]. The inhibitory effect of Quercimeritrin on α-amylase was significantly higher than that of Hyperoside, and it can be seen from the number of OH on the AB ring that the increase in OH number is beneficial to enhancing the inhibitory effect of compounds on α-amylase. Similarly, in the study of the structure law of irisolidone, irigenin, and iridin A on α-amylase inhibition, it was found that when the number ofOH groups in the A ring and B ring increased, the inhibitory effect of the compound on α-amylase was enhanced [30]. The inhibitory effect of Apigetrin and Oroxin A on α-amylase was significantly higher than that of Quercimeritrin and Hyperoside, and it can be seen from the structure that OH at the A6 position can significantly increase the affinity of the compound for α-amylase. By exploring the inhibitory effect of flavonoids on porcine pancreatic α-amylase, studies found that the presence ofOH groups at positions 6 and 7 of the A ring and position 4 of the B ring enhanced the inhibitory effect of α-amylase, whereas the glycan moieties on the flavonoid structure reduced their ability to inhibit α-amylase [30,31]. The inhibitory activity of Catechin gallate and Epicatechin gallate is higher among many compounds, indicating that the introduction of galloyl groups enhances the α-amylase inhibitory activity of compounds, and cis- and trans-isomers have an effect on the inhibitory properties of compounds. The same conclusion was reached in the study by Desseaux, where the presence of gallic acyl groups led to a modest reduction in the value of the inhibitory constant (*K*i) [32].

Hyperoside, Trifolin, Avicularin, and Quercitrin all introduced glycosidyl groups at position C3, but their inhibitory activity on α-glucosidase was lower than that of other compounds at position C3OH, suggesting that the presence ofOH groups at C3 is more favorable than the presence of glycoside groups, and glycosylation at the A7 position is conducive to the enhancement of enzyme inhibitory activity. In addition, the inhibitory activity (%) of these three compounds on α-glucosidase was Hyperoside > Quercitrin > Avicularin > Trifolin, indicating that OH at the B5′ position was beneficial to the enhancement of inhibitory activity, and galactoside in the C3 position was more favorable than the presence of rhamnose and arabinofuranoside. When Silva et al. studied the flavonoids isolated in *Saccharomyces cerevisiae*, they found that the presence of theOH group at the 3-position of the C ring was more conducive to α-glucosidase inhibition activity [33]. In addition, the presence of theOH group at the B5′ position seemed to increase the inhibitory effect of α-glucosidase [34]. Flores-Bocanegra et al. also found that the type of glycoside group seemed to affect inhibitory activity, as glucoside isoquercetin (glucoside) was more active than hypericin (galactoside) and quercetin (rhamnoside) [19].

Among the 12 compounds, the glucosidase inhibitory activity of Apigetrin, Quercetagitrin, Quercimeritrin, Catechin gallate, and Epicatechin gallate was higher than that of acarbose, but only Apigetrin and Quercimeritrin showed selective inhibition of glucosidase. According to the purpose of our experiment, Quercimeritrin was mainly used to inhibit glucosidase, and Quercimeritrin was finally selected as a selective inhibitor for a follow-up study.

In general, for flavonoid glycosides, glycosylation at the A7 position (especially the substitution of glucoside groups) can significantly enhance the inhibitory effect of the compounds on α-glucosidase. The OH group at the A6 position can enhance the inhibition of α-amylase and weaken the inhibitory effect of α-glucosidase. TheOH groups at the B4′ and B5′ positions are closely related to the inhibition of α-glucosidase, but the OH group at the B4′ position is not conducive to the inhibition of amylase, whereas the OH group at the B5′ position increases the inhibition activity of α-amylase. Among many compounds, quercetin is the most suitable compound for selective inhibition of α-glucosidase. Its inhibitory effect on α-amylase is much lower than that of acarbose, and it has a strong inhibitory activity for α-glucosidase.

### 3.2. Inhibition Kinetics of Compounds on Starch-Digesting Enzymes

The findings demonstrated that Quercimeritrin had a stronger inhibitory impact on α-glucosidase than it did on α-amylase, demonstrating that it had a selective inhibitory effect on α-glucosidase. In addition, the IC_50_ value of Quercimeritrin on α-glucosidase was much lower than that of acarbose, and the IC_50_ value of Quercimeritrin on α-amylase was much higher than that of acarbose, indicating that Quercimeritrin can effectively reduce the side effects of existing glucosidase inhibitors compared with acarbose while controlling postprandial blood glucose [28]. To further determine the kinetic mechanism of Quercimeritrin inhibiting α-glucosidase and α-amylase, the Lineweaver–Burk diagram was used for kinetic analysis. In the Lineweaver–Burk diagram (Figure 2e), all lines intersect in the negative direction of the *Y*-axis, *V*max remains constant, and *K*m decreases with increasing Quercimeritrin concentration. Hence, it is assumed that Quercimeritrin inhibits α-glucosidase by competitive inhibition [35]. This indicates that Quercimeritrin inhibits glucosidase by interfering with the amino acid residues in the enzyme activity pocket, causing steric hindrance and reducing enzyme activity. Other flavonoid glycosides, such as PG3G and M3A, have also been reported to inhibit α-glucosidase competitively [26]. Interestingly, Quercimeritrin also showed competitive inhibition in the inhibition mode of amylase, but its *K*m value for α-amylase was much higher than that for α-glucosidase, indicating that Quercimeritrin had a higher affinity for α-glucosidase. This finding was also consistent with the results of IC_50_ experiments [36].

### 3.3. Enzyme and Quercimeritrin Affinity Verification

The varied inhibitory effects of Quercimeritrin imply that these compounds have different binding affinities for the two groups of starch-digesting enzymes. In this paper, the tryptophan fluorescence quenching method was used to characterize the degree of affinity between compounds and enzymes [37]. First, it is necessary to determine whether the compound interacts with the two enzymes. The endogenous fluorescence caused by Trp and Tyr residues at around 340 nm is what gives α-glucosidase and α-amylase their highest fluorescence intensity. Therefore, the binding degree can be determined by measuring the effect of the compound on the emission spectra of the two enzymes [38]. Figure 3 shows that with the increase in Quercimeritrin concentration, the fluorescence intensity of the two enzymes decreases significantly, indicating that Quercimeritrin can quench the intrinsic fluorescence of the two enzymes and bind to them. However, by comparing the effects of Quercimeritrin with the same concentration on enzyme fluorescence, it can be seen that Quercimeritrin has a greater effect on the fluorescence quenching value of α-glucosidase, which also indicates that Quercimeritrin is more compatible with α-glucosidase.

The quenching mechanism of Quercimeritrin was further investigated to better comprehend how it interacts with the two enzymes [23]. By determining the quenching constant, it is possible to discriminate between dynamic and static protein fluorescence quenching caused by compounds [39]. Because the quenching agent reacts with fluorescent groups to create nonfluorescent substances, static quenching occurs. With the increase in temperature, the binding of the complex is unstable, so the quenching constant decreases. The drop in fluorescence intensity brought on by the interaction of the quencher and the fluorescence group is what causes dynamic quenching. When the collision between the quencher and the fluorescence group intensifies as the temperature rises, the quenching constant rises [35]. The quenching mechanism of the enzyme was further studied using the Stern–Volmer equation:(2)$$\frac{F_{0}}{F}=1+K_{q}\tau 0\left[Q\right]=1+K_{sv}[Q],$$
where *K*_q_ is the bimolecular quenching constant, *F*_0_ represents the fluorescence intensity of starch-digesting enzymes in the absence of inhibitors, *F* represents the fluorescence intensity of starch-digesting enzymes in the presence of inhibitors, and τ0 is the lifetime of the fluorophore in protein (about 10^−8^ s). [Q] is the concentration of Quercimeritrin as a quenching agent, and *K*_sv_ is the Stern–Volmer quenching constant. 

The Stern–Volmer diagrams (Figure 3c–f) are used to calculate *K*_sv_ at the three temperatures (300.15, 305.15, and 310.15 K) shown in Appendix A. *K*_sv_ displays the inhibition of the quenching agent on the biofluorescence of the receptor [40]. As can be seen from Appendix A, with the increase in temperature, the *K*_sv_ value increases from 0.17 to 0.21 × 10^5^ L/mol (Quercimeritrin-α-glucosidase) and 0.13 to 0.18 × 10^5^ L/mol (Quercimeritrin-α-amylase). The higher the *K*_sv_ value, the stronger the quenching inhibition of α-glucosidase, indicating that Quercimeritrin is more tightly bound to α-glucosidase and has stronger quenching inhibition than α-amylase. Subsequently, according to the *K*_sv_ value, the *K*_q_ value can be calculated using Equation (2). The results of Appendix A show that the *K*_q_ value is much greater than 2.0 × 10^10^ L/mol (dynamic quenching maximum collision constant). Thus, it can be determined that the quenching caused by Quercimeritrin is due to the formation of a ground state complex rather than dynamic collisions. The static quenching mechanism is the main reason to control the fluorescence quenching process [41].

### 3.4. Binding Sites and Binding Constants

In order to further determine the binding constant (*K_a_*) and number of binding sites (n) of the inhibitor binding to the enzyme, the following formula is used to calculate:(3)$$log\frac{\left(F_{0}-F\right)}{F}=logK_{a}+nlog[Q].$$
where *K_a_* is the binding constant, *n* is the number of binding sites, *F*_0_ represents the fluorescence intensity of starch-digesting enzymes in the absence of inhibitors, *F* represents the fluorescence intensity of starch-digesting enzymes in the presence of inhibitors, and [Q] is the concentration of the Quercimeritrin.

According to the double logarithmic equation (Equation (3)), the binding constant (*K_a_*) and the number of binding sites (*n*) were calculated (Appendix A), indicating that the binding constant of Quercimeritrin with α-glucosidase (0.41 ± 0.06 × 10^5^ /M, 310.15 K) was greater than that with α-amylase (0.26 ± 0.01 × 10^5^ /M, 310.15 K), showing that the binding forces of Quercimeritrin and α-amylase were stronger than that of α-amylase [42]. Meanwhile, *n* is approximately equal to 1, indicating that Quercimeritrin has only one binding site for both α-amylase and α-glucosidase. In addition, the fluorescence spectrum of Quercimeritrin and the two enzymes did not exhibit a redshift at the 340 nm emission peak (Figure 3a,b), indicating that the flavonoid compounds were located close to tryptophan residues and could cause fluorescence quenching without altering the structure of the enzymes [42]. Similar conclusions were also obtained in the study on the mechanism of phloretin’s inhibition of α-glucosidase. The maximum emission wavelength of α-glucosidase was redshifted, and the spatial structure of the enzyme was changed by the inhibitor, leading to a more polar and exposed microenvironment of amino acid residue [43].

### 3.5. Thermodynamic Parameters and Binding Forces

By calculating the thermodynamic parameters of the quenching process, the binding affinity and the main force of the reaction process were evaluated [44]. The thermodynamic parameters are calculated using the Van ’t Hoff equation:(4)$$\ln K_{a}=-\frac{△H}{RT}+\frac{△S}{R}.$$
(5)△G=△H−T△S.
where *K_a_* is the binding constant, R is the universal gas constant, and T represents the test temperature (300.15 K, 305.15 K, 310.15 K). The thermodynamic parameters ΔH, ΔS, and ΔG represent enthalpy change, entropy change, and free energy change, respectively. The thermodynamic parameters of Quercimeritrin and the two enzymes at different temperatures were calculated (Appendix A), where ΔG < 0 indicates that the binding of Quercimeritrin to the two enzymes is a spontaneous process. The ΔG value of the interaction between Quercimeritrin and α-glucosidase is smaller than that between Quercimeritrin and α-amylase, which also indicates that Quercimeritrin is more likely to interact with α-glucosidase than α-amylase. The interaction forces in the reaction process can be roughly divided into four types: hydrophobic interaction, hydrogen bonding, electrostatic interaction, and van der Waals interaction [45]. According to the theory, if the thermodynamic calculation gets a ΔH < 0, ΔS < 0 indicates that the main binding force of the inhibitor and enzyme is hydrophobic interaction, while ΔH < 0, ΔS < 0 indicates that the inhibitor–enzyme interaction is mainly achieved via hydrogen bonds and van der Waals forces [44,46]. From Appendix A, it can be observed that both ΔH < 0 and ΔS < 0 values for the binding of Quercimeritrin with the two enzymes are negative. This indicates that the main forces of the binding process are hydrogen bonding and van der Waals forces, which are related to multiple hydroxyl groups in Quercimeritrin molecular structure and protein molecules. Additionally, it also suggests that the binding process is primarily thermally driven, resulting in an exothermic reaction [47].

### 3.6. Synchronous Fluorescence Spectra

Synchronous fluorescence spectra can be used to reflect the changes of conformation and microenvironment near protein fluorophores. When Δλ is set at 15 nm and 60 nm, it can represent the changes of Tyr and Trp residues, respectively. The synchronous fluorescence quenching ratio (RSFQ) is calculated using the following formula:$$RSQF=1-F/F_{0}$$

As shown in the Appendix A, with the increase in Quercimeritrin concentration, the synchronous fluorescence intensity of α-glucosidase at Δλ = 15 nm and Δλ = 60 nm gradually decreased, and the maximum absorption peak of Trp residue showed a significant redshift, while the maximum absorption peak of Tyr residue remains relatively unchanged. This indicates that the inhibitor interacts with α-glucosidase and changes the polarity of the microenvironment near Trp residues but has little effect on the hydrophobic microenvironment near Tyr. The RSFQ graph reflects the extent of protein quenching performed by the inhibitor, further confirming that the inhibitor indeed has a stronger impact on Trp [44]. By comparing the RSFQ diagram of the two enzymes, it can be observed that the quenching degree of α-glucosidase by the inhibitor is higher than that of α-amylase regardless of Trp or Tyr.

In general, the affinity of Quercimeritrin for α-glucosidase was much greater than that for α-amylase, which was consistent with the results of the inhibition experiments of the two enzymes in the previous paper, which strongly supported the selective inhibition of the Quercimeritrin for the two digestive enzymes. The high binding affinity of Quercimeritrin to α-glucosidase may be due to the differences in the chemical properties of the activity pockets of the two enzymes [48].

### 3.7. Molecular Docking and Molecular Dynamics Simulation (MD)

Quercimeritrin was characterized by inhibition kinetics and fluorescence quenching experiments to have a stronger affinity with α-glucosidase than with α-amylase, thus producing a stronger inhibitory effect. Therefore, molecular docking and MD methods were applied to explore further the interaction mechanism of Quercimeritrin with α-glucosidase and compare it with α-amylase to find the difference in the mechanism. First, using molecular modeling, the inhibitory mechanism of Quercimeritrin on the α-glucosidase and α-amylase was investigated. This mechanism is mainly based on steric hindrance, in which inhibitors dock in the active pocket of α-glucosidases and interact with the amino acid residues of the active pocket, resulting in steric hindrance to the binding of other substrates [49]. The calculation results, in this case, support the thermodynamic finding that the glycosyl scaffold of the compound is positioned toward the core of the binding pocket and interacts closely with important residues at the active site, thereby blocking the catalytic process to help produce inhibition (Figure 3b). From the overall score results (Appendix A), the intermolecular energy is predicted to be −81.17 kcal/mol, in which the hydrogen bond and van der Waals force energies are −53.93 kcal/mol, and the electrostatic energy is −27.24 kcal/mol, indicating that hydrogen bonding dominates the intermolecular binding process.

Figure 4a,b shows the active pocket structure, the shape of the Quercimeritrin molecule, and the associated amino acid residues, and Figure 4c shows the interaction of the compound with α-glucosidase to better show how Quercimeritrin and α-glucosidase interact. The results showed that Quercimeritrin formed a hydrophobic interaction with some amino acid residues after entering the catalytic site of α-glucosidase, as shown in Figure 4b. In addition, there were hydrogen bonds around Quercimeritrin and amino acid residues ARG202, ASP203, ASP327, TRP406, ASP443, and ASP542. These hydrogen bonds show another important interaction between ligands and proteins. The docking of Quercimeritrin with α-amylase was also performed. The results (Appendix A) showed that Quercimeritrin could dock in the active pocket of α-amylase, but its docking score was much lower than that of α-glucosidase (Appendix A). In conclusion, the interaction between Quercimeritrin and α-glucosidase was strong, which enabled it to firmly occupy the active pocket of the enzyme and reduce the enzyme activity through steric hindrance, providing an inhibitory effect.

Subsequently, MD simulation was used to analyze the substance (Quercimeritrin) that exhibited the strongest inhibitory activity with the α-glucosidase complex and the free α-glucosidase. The dynamic stability and structural alterations of the complex systems were assessed by looking at the radius of gyration (*R*g), root-mean-square fluctuation (RMSF), and root-mean-square deviation (RMSD) of the protein backbone. All three of those compound systems’ RMSD values fluctuated around 1.2, indicating a comparatively stable system (Figure 5a). The majority of the residues had RMSF values under 2, indicating similar internal flexibility and fairly stable structures (Figure 5b). When compared with the free α-glucosidase, there was a significant fluctuation in the residues of ARG202–ASP203, ILE364–ASN372, ASP443–ASP452, LYS543–ALA545, and SER825–ASN832 residues for the complex of Quercimeritrin with the enzyme. These results suggested that the amino acids had more adaptable structures or that these regions were important for carrying out certain protein activities. This part of the wave peptide is exactly consistent with the amino acid of the above molecule to produce hydrogen bonds, indicating that Quercimeritrin interacts with α-glucosidase, forming an enzyme–substrate complex that causes inhibition. The term “radius of gyration” (*R*g) refers to how compact a protein structure is. A more stable protein is indicated by a lower value. It was discovered that Quercimeritrin and α-glucosidase complex had a smaller *R*g than free α-glucosidase. Because Quercimeritrin formed a more stable system and had a lower *R*g value, it was more likely to bind with α-glucosidase and demonstrate substantial inhibitory action (Figure 5d).

In summary, molecular docking and MD experiments have demonstrated that Quercimeritrin can dock in the active pockets of enzymes and generate intermolecular forces with some amino acid residues. These interactions allow enzymes and Quercimeritrin to form more stable and compact structures, preventing other substrates from binding to enzymes and, thus, inhibiting enzyme activity.

### 3.8. Quercimeritrin Effect on Postprandial Blood Glucose in Mice

We subsequently examined the impact of Quercimeritrin on blood glucose levels in an in vivo starch load test to confirm if the inhibitory effect of Quercimeritrin on α-glucosidase was the same as that of in vitro investigations. α-Amylase hydrolyzes starch into maltose, and α-glucosidase hydrolyzes maltose into glucose, causing postprandial blood sugar to rise. As a result, blood glucose levels following oral maltose indirectly reflect the activity of α-glucosidase and/or α-amylase. According to Figure 6a, blood sugar levels in starch-fed mice increased gradually over the first 0.5 h, peaking at 1 h, before decreasing and eventually approaching fasting blood glucose levels. The hypoglycemic ability of the LD group was similar to that of the acarbose group, whereas the postprandial blood glucose level of the HD group of treated mice was significantly lower than that of the control and acarbose groups (Figure 6a).

The changes in blood glucose in mice were expressed as an area under the glucose curve (AUC). As shown in Figure 6b, acarbose (100 mg/kg) reduced the glucose AUC of diabetic mice by 20.31% (43.0 mmol/(L*h)), 19.0% (43.7 mmol/(L*h)) for Quercimeritrin (100 mg/kg), and 24.5% (40.7 mmol/(L*h)) for Quercimeritrin (200 mg/kg). These results indicate that Quercimeritrin can significantly inhibit postprandial hyperglycemia in diabetic mice at the in vivo level, which is consistent with the results of in vitro inhibition experiments.

## 4. Conclusions

In this study, we investigated the inhibitory activities of 12 flavonoid glycosides on α-amylase and α-glucosidase and found that some flavonoid glycosides with specific structures could meet the need for selective inhibition of the two starch-digesting enzymes. The glycoside group at A7 and the OH groups at B5′ and B4′ are conducive to the inhibition of α-glucosidase. TheOH groups at A6, C3, B4′, and B5′ are favorable for the inhibition of α-amylase. In addition, in vitro and in vivo experiments jointly verified that Quercimeritrin can be used as a highly effective selective inhibitor of α-glucosidase. Its selective inhibition mechanism is as follows: Quercimeritrin has a higher affinity for α-glucosidase and interacts with amino acid residues in the active pocket of α-glucosidase through noncovalent bonds (mainly hydrogen bonds), resulting in conformational changes of the enzyme to form an enzyme–Quercimeritrin complex, which plays an inhibitory role. In the future, in addition to developing more effective selective inhibitors, we need to determine how to apply selective inhibitors of α-glucosidase to achieve slow but complete digestion of starch, eliminating the side effects of existing inhibitors while smoothly controlling postprandial blood glucose.

## Figures and Tables

**Figure 1 foods-12-03415-f001:**
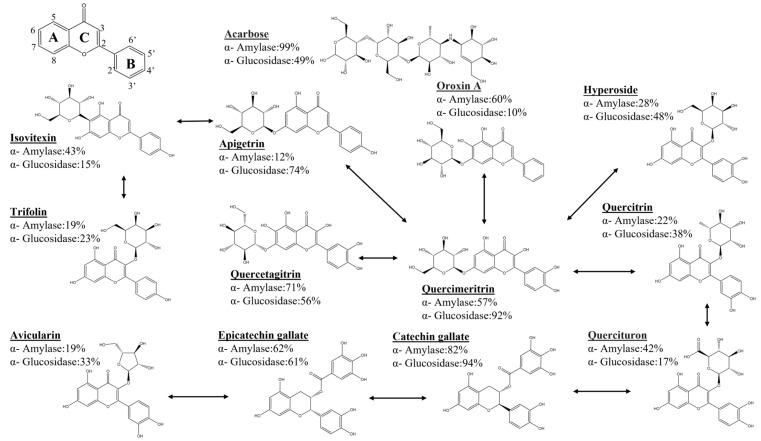
Selective inhibition of starch-digesting enzymes by flavonoid glycosides. The inhibitory effects of each compound on two kinds of starch-digesting enzymes at 500 µM are shown in the figure. The arrows show the relationship between the structures of the compounds, and compound names in bold font.

**Figure 2 foods-12-03415-f002:**
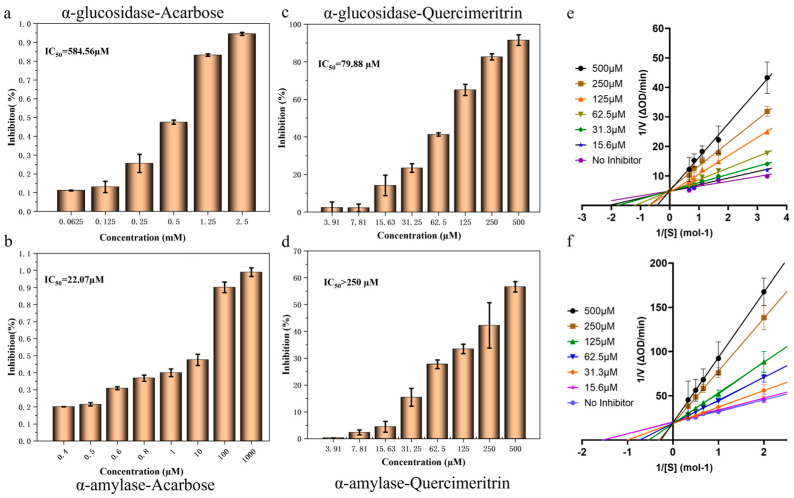
Inhibitory effect of acarbose and Quercimeritrin on α-glucosidase and αamylase. (**a**) IC_50_ value of acarbose inhibited α-glucosidase, (**b**) IC_50_ value of acarbose inhibited α-amylase, (**c**) IC_50_ value of Quercimeritrin inhibited α-glucosidase, (**d**) IC_50_ value of Quercimeritrin inhibited α-amylase, (**e**) Lineweaver−Burk plots of Quercimeritrin on α-glucosidase, and (**f**) Lineweaver−Burk plots of Quercimeritrin on α-amylase.

**Figure 3 foods-12-03415-f003:**
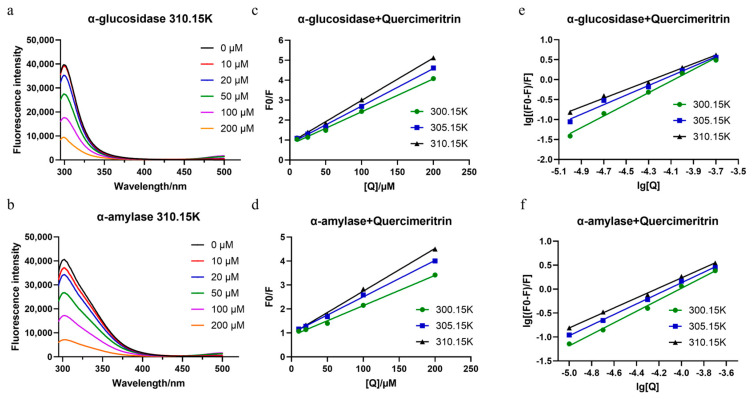
(**a**,**b**) Fluorescence spectra of α-glucosidase and α-amylase in the presence of Quercimeritrin at various concentrations, (**c**,**d**) the Stern−Volmer plots of α-glucosidase and α-amylase in the presence of Quercimeritrin at different temperatures, and (**e**,**f**) logarithmic plots of α-glucosidase and α-amylase in the presence of Quercimeritrin at gradient temperatures.

**Figure 4 foods-12-03415-f004:**
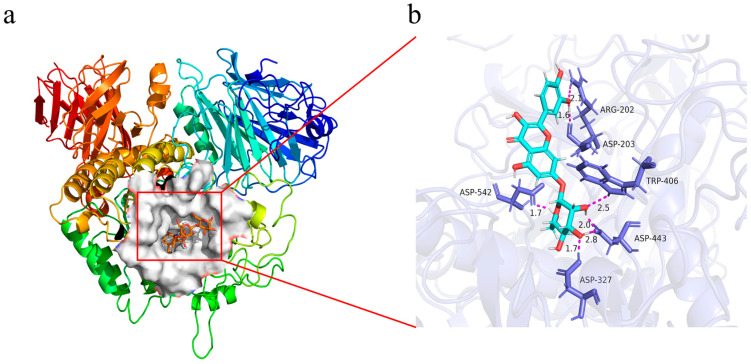
The way the compound binds to glucosidase. (**a**) An overall diagram of the hydrophobic pocket and active site interaction of the compound with α-glucosidase, and (**b**) three-dimensional binding of the compound with α-glucosidase. Labeled key residues, hydrogen bonds formed between ligands, and key residues (purple) are shown as dotted or solid lines. Blue compounds represent inhibitors, and purple represents amino acid residues.

**Figure 5 foods-12-03415-f005:**
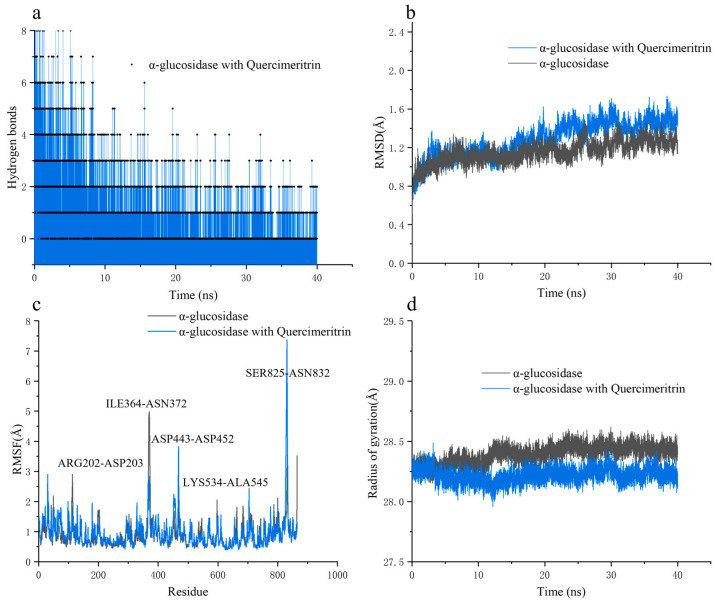
Molecular dynamics of Quercimeritrin and α-glucosidase. (**a**–**d**) represent the hydrogen bond change spectrum, RMSD change spectrum, RMSF value, and radius of gyration of quercetin and α-glucosidase, respectively.

**Figure 6 foods-12-03415-f006:**
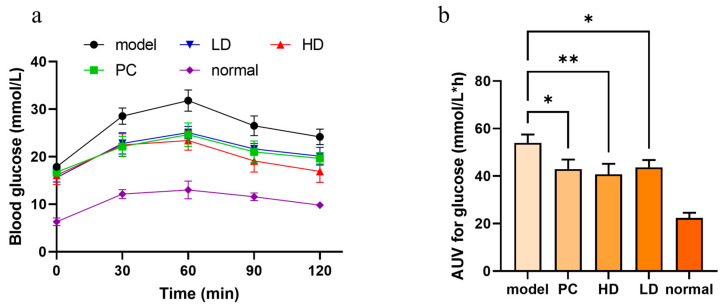
Blood glucose levels in diabetic mice after administration. (**a**) Blood glucose concentration in mice after oral administration of starch. normal saline (model group, model), 100 mg/kg acarbose (positive control group, PC), 100 mg/kg Quercimeritrin (low -dose group, LD), 200 mg/kg Quercimeritrin (high-dose group, HD), and C57BL/6J was used as blank controls (normal). (**b**) The corresponding AUC after oral administration of starch in mice. Data are represented as means ± SEM (*n* = 10), * *p* < 0.05 vs. model, ** *p* < 0.01 vs. model.

## Data Availability

The datasets generated for this study are available on request to the corresponding author.

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
