# Peer review of "Inhibitory Mechanism of Quercimeritrin as a Novel α-Glucosidase Selective Inhibitor"

_foods, 2023, doi:10.3390/foods12183415_

Round 1
Reviewer 1 Report
With the knowledge that the control of postprandial blood glucose is a key step in the treatment of obesity and diabetes, the inhibition of carbohydrate enzymes could be a potential target for lowering the risk factors of diabetic complications. The paper “Inhibitory mechanism of Quercimeritrin as a novel α-glucosidase selective inhibitor” assessed in vitro the enzyme activity inhibition potential of twelve flavonoid glycosides, showing that quercimeritrin was the best inhibitor of α-glucosidase. The selective inhibition mechanism for α-glucosidase and α-amylase of this glycoside was further in silico analyzed, while in vivo experiments exposed a good blood glucose control, concluding that quercimeritrin could be effectively used to ameliorate postprandial blood glucose levels.
In summary, the title stresses the value of the study; the abstract includes sufficient information to stand alone; the introduction summarizes the topic current state and knowledge in the field and explains why the experiment was needed; the methods are distinctly described with adequate details; the results are accurately presented, with relevant data given in tables and figures; in the discussion chapter the findings of the study are logically explained and compared to other findings in the field; the conclusions support and clearly summarize the experiment.
The topic is very important and the manuscript provides a comprehensive analysis of the subject. I have no hesitation in recommending this manuscript; the addition of the following references in Introduction or Discussion may improve the quality of the paper:
- extracts of Astragalus creticus, containing quercimeritrin among other secondary metabolites, demonstrated antioxidant potential and good inhibition of α-glucosidase enzyme (PMID: 34275786)
- recent studies showed that other flavones, tricetin and genkwanin, exhibited α-glucosidase and α-amylase inhibitory activities, respectively (doi: 10.1080/14786419.2018.1446009; doi: 10.1016/j.biopha.2023.115159), while linarin could prevent the action of α-glucosidase and α-amylase enzymes (doi: 10.1080/07391102.2021.1882340)
Author Response
Comments 1: the addition of the following references in Introduction or Discussion may improve the quality of the paper:
- extracts of Astragalus creticus, containing quercimeritrin among other secondary metabolites, demonstrated antioxidant potential and good inhibition of α-glucosidase enzyme (PMID: 34275786)
- recent studies showed that other flavones, tricetin and genkwanin, exhibited α-glucosidase and α-amylase inhibitory activities, respectively (doi: 10.1080/14786419.2018.1446009; doi: 10.1016/j.biopha.2023.115159), while linarin could prevent the action of α-glucosidase and α-amylase enzymes (doi: 10.1080/07391102.2021.1882340)
Response 1: Thank you for pointing this out. The citations of these four papers can really improve the quality of the paper, We have added and marked in red in the article. (-line 64-71)

Reviewer 2 Report
This is an interesting paper covering key aspects of the inhibition mechanism of selected flavonoids, as well as corroborating in vivo the effective of Quercimeritrin as a novel α-glucosidase selective inhibitor. It is well written, clear, and consistent. No major deficiencies were identified.
I suggest considering these changes:
Line 195: “low-dose” group must be LD, while “high-dose” group HD.
Figure 6: please identify the abbreviations in the caption. It is a bit incommode go back in the manuscript to understand each one.
Author Response
Comments 1: Line 195: “low-dose” group must be LD, while “high-dose” group HD.
Response 1: Thank you for pointing this out. We agree with this comment. We have revised it in the original text. -line 200-201
Comments 2: Figure 6: please identify the abbreviations in the caption. It is a bit incommode go back in the manuscript to understand each one.
Response 2: Thanks for your comments, it's a little inconvenient to go back to the original draft and understand each abbreviation, so we've added the corresponding comment for the abbreviation in Figure 6. – line 531-533

Reviewer 3 Report
In the article (foods-2578693) entitled “Inhibitory mechanism of Quercimeritrin as a novel α-glucosidase selective inhibitor”, Quercimeritrin was selected as the best selective inhibitor of α-glucosidase through in vitro enzyme activity inhibition experiments. It could be used as a new selective inhibitor of α-glucosidase. The selective inhibition mechanism of Quercimeritrin on the two starch-digesting enzymes was further explored, and it was confirmed that Quercimeritrin had a strong binding affinity for α-glucosidase and occupied the binding pocket of α-glucosidase through non-covalent binding. Therefore, the results provide insights into how flavone aglycones can be used to effectively control the rate of digestion to improve postprandial blood glucose levels.
Please follow instructions for authors.
Several abbreviations should be written in full words.
Please change IC50 to IC50.
Please change low-dose group, HD to low-dose group, LD in all places.
Please change high-dose group, LD to high-dose group, HD in all places.
Please add more information about statistical analysis.
“P values” should be given in the results and figures when it is needed.
Please describe figures appropriately and add proper captions.
Please compare with more similar studies.
Please follow instructions for authors, especially in citations and references.
For example, for citations:
Line 29 diabetic complications (Bello et al., 2014) change to [1].
The same is for others.
Moderate editing of English language is required.
Author Response
Please see the attachment of the author's response.

Reviewer 4 Report
Comments to Manuscript ID: foods-2578693.
The article entitled as “Inhibitory mechanism of Quercimeritrin as a novel α-glucosidase selective inhibitor”.
The inhibitory effect of Quercimeritrin on α-amylase and α-glucosidase has been studied. Interesting features have been discovered.
The article contains inaccuracies in the presentation and discussion of the results, which are listed in Comments to authors.
With the conclusions of the authors in section 3.1. (Structural properties of compounds that selectively inhibit starch-digesting enzymes) can be argued.
I apologize that those sections 3.5. (Thermodynamic parameters and binding forces) and 3.7. (Molecular docking and Molecular dynamics simulation (MD)) have not been peer-reviewed as they are not within the scope of my research.
Author Response
Comments 1: With the conclusions of the authors in section 3.1. (Structural properties of compounds that selectively inhibit starch-digesting enzymes) can be argued.
Response 1: Thank you for your excellent comments. This part of the structure-activity relationship is indeed controversial in a large number of literatures, and I think it should be related to the type of compound. The structural characteristics of selective inhibition of starch-digesting enzymes in Section 3.1 were inferred from the results obtained in this experiment and compared with the conclusions reported in a large number of literatures, and the results obtained have also been reported in other types of compounds.
Comments 2: I apologize that those sections 3.5. (Thermodynamic parameters and binding forces) and 3.7. (Molecular docking and Molecular dynamics simulation (MD)) have not been peer-reviewed as they are not within the scope of my research.
Response 2: Thanks for your comments, we have reviewed the discussion section of sections 3.5. (Thermodynamic parameters and binding forces) and 3.7. (Molecular docking and Molecular dynamics simulation (MD)) in detail